# The Effects of Ultrasound Treatment of Graphite on the Reversibility of the (De)Intercalation of an Anion from Aqueous Electrolyte Solution

**DOI:** 10.3390/nano12223932

**Published:** 2022-11-08

**Authors:** Ghulam Abbas, Zahid Ali Zafar, Farjana J. Sonia, Karel Knížek, Jana Houdková, Petr Jiříček, Martin Kalbáč, Jiří Červenka, Otakar Frank

**Affiliations:** 1J. Heyrovsky Institute of Physical Chemistry, Czech Academy of Sciences, Dolejskova 2155/3, 183 23 Prague, Czech Republic; 2Department of Physical Chemistry and Macromolecular Chemistry, Faculty of Science, Charles University in Prague, Hlavova 2030, 128 43 Prague, Czech Republic; 3FZU—Institute of Physics of the Czech Academy of Sciences, Cukrovarnicka 10/112, 162 00 Prague, Czech Republic

**Keywords:** ultrasonication, graphite, intercalation, in situ Raman spectroelectrochemistry, operando XRD, aqueous electrolyte

## Abstract

Low cycling stability is one of the most crucial issues in rechargeable batteries. Herein, we study the effects of a simple ultrasound treatment of graphite for the reversible (de)intercalation of a ClO_4_^−^ anion from a 2.4 M Al(ClO_4_)_3_ aqueous solution. We demonstrate that the ultrasound-treated graphite offers the improved reversibility of the ClO_4_^−^ anion (de)intercalation compared with the untreated samples. The ex situ and in situ Raman spectroelectrochemistry and X-ray diffraction analysis of the ultrasound-treated materials shows no change in the interlayer spacing, a mild increase in the stacking order, and a large increase in the amount of defects in the lattice accompanied by a decrease in the lateral crystallite size. The smaller flakes of the ultrasonicated natural graphite facilitate the improved reversibility of the ClO_4_^−^ anion electrochemical (de)intercalation and a more stable electrochemical performance with a cycle life of over 300 cycles.

## 1. Introduction

Overwhelming energy demand has propelled paramount interest in discovering the best ways to store energy in an environmentally friendly and sustainable manner [1]. For the development of more efficient and stable energy storage technologies, it is vital to explore different possible electrochemical charge storage mechanisms [2]. Graphite is one of the most studied electrode materials for batteries and supercapacitors, and has been extensively used in many lithium-ion, metal-ion, and nonmetal-ion energy storage devices [3,4]. The advantages of graphite are its high abundance, low cost, and versatility. Graphite allows the formation of many different types of graphite intercalation compounds (GICs) using both cations and anions [5,6]. For this reason, graphite is one of the most explored electrode materials in dual-ion batteries (DIBs). 

DIBs are the trending contender in energy storage systems because they offer a wide variety of viable intercalation chemistries. Graphite-based electrodes are also favorable for DIBs because of the mechanical integrity of GICs. Graphite can accommodate even large ions and provide high cutoff voltages (e.g., 2.0–5.2 V vs. K/K^+^ or 2.4–3.7 V vs. Al/Al^3+^) and specific capacities (50–300 mAh g^−1^) in organic and ionic liquid electrolytes [7]. GICs can be formed using metal cations, such as Li^+^, Na^+^, K^+^, Zn^2+^, Al^3+^, and anions such as bis(trifluoromethanesulfonyl) imide TFSI^−^, hexafluorophosphate PF_6_^−^, tetrafluoroborate BF_4_^−^, tetrachloroaluminate AlCl_4_^−^, and perchlorate ClO_4_^−^ [8,9,10,11,12,13,14,15,16]. Even though graphite-based DIBs based on ionic liquid and organic electrolytes are promising in terms of stable and reversible electrochemical performance, they have demerit points, such as flammability, high cost, environmental issues, or unsuitability for an open-air environment. These issues severely hinder their applicability across a wide variety of technology fields [17,18].

Aqueous electrolytes can potentially overcome the issues of organic and ionic liquid electrolytes in DIBs due to their inherent nonflammability, high ionic conductivity, and easy-to-handle nature in an open-air environment [19,20]. However, aqueous electrolytes suffer from small electrochemical potential windows of hydrogen and oxygen evolution reactions (~1.23 V) [21]. Highly concentrated aqueous electrolytes have recently enabled enlarging the electrochemical potential window by suppressing free water and reducing electrolyte decomposition [22]. Increased salt concentration in aqueous electrolytes has also enabled researchers to obtain lower-stage GICs and improve the reversibility of ionic (de)intercalation [23].

One of the highest electrochemical stabilities has been reported in highly concentrated Al(ClO_4_)_3_ and Zn(ClO_4_)_2_ aqueous electrolytes that offer an electrochemical window in the range of 3–4 V and the facile intercalation of the ClO_4_^−^ anion in graphite [14,23]. However, the cycling stability and reversible ionic de(intercalation) in graphite during the discharge process are still key issues in ClO_4_^−^-based electrolytes, limiting the cycling stability of perchlorate-based aqueous DIBs [14]. 

Over the years, researchers have investigated several different approaches for improving the reversibility of graphite intercalation in aqueous electrolytes [24,25,26]. Ultrasound treatment has changed the structural properties of graphite by inducing defects in the lattice structure and reducing its crystallite size [24,25,26]. The impact of the crystallite domain size (*L*_a_) and degree of graphitization (*g*) on anion intercalation in graphite has been reported by Heckmann et al. [27], and also in our previous publication [28]. It has been found that a smaller crystallite domain size enhances ionic transport kinetics, which in turn improves the reversibility during the electrochemical (dis)charge process [29]. Generally, the reversibility of the intercalation process and kinetics at the electrode surface can be monitored by cyclic voltammetry (CV). However, to gain deeper insights into electrochemical ionic intercalation and to understand its correlation with the incurred structural modifications, various in situ studies featuring techniques such as X-ray diffraction (XRD), transmission electron microscopy (TEM), Raman spectroscopy, and X-ray photoelectron spectroscopy (XPS) are required [30,31]. Ostwald et al. performed in situ XPS to investigate the surface evolution of a graphite electrode material during the intercalation process in a Li-ion battery [32]. However, due to depth limitations, in situ XPS can only provide information about the first few nanometers from the surface of the material. TEM can provide atomically resolved pictures of the insertion/extraction process of the reactive ion [33], but the experiments are extremely demanding and cannot be routinely performed. Therefore, in situ Raman spectroelectrochemistry (SEC) and operando XRD are the most prominent tools that have been utilized to explore structural changes during the ion storage mechanism in detail [34,35]. Operando XRD reveals the lattice changes (contraction/expansion) in bulk during the electrochemical ionic intercalation process [36,37,38,39]. The operando XRD can be complemented by more surface-sensitive in situ Raman SEC, to provide information from the first tens to hundreds of nanometers depending on the studied material. It can also provide information about the reversibility of the charge/discharge process and the disorder induced in the material [23,28]. 

In this study, we report the effects of the mild ultrasound treatment of graphite on the reversibility of the (de)intercalation of the ClO_4_^−^ anion using a 2.4 M Al(ClO_4_)_3_ aqueous solution. The intercalation process is characterized by electrochemistry (galvanostatic charge/discharge and CV) and by various in situ and ex situ techniques, including Raman spectroscopy, SEC, XRD, and XPS. The obtained results provide insight into the ClO_4_^−^ anion intercalation mechanism in graphite, demonstrating a positive correlation between the (de)intercalation reversibility and the decrease in the lateral crystallite size by the ultrasound treatment. 

## 2. Material and Methods

### 2.1. Material Preparation

A total of 200 mg of natural graphite (NG) (Nacional de Grafite, Ltda, Itapecerica, Brazil) was added into 60 mL of N-methyl-2-pyrrolidone (NMP) (Roth, Karlsruhe, Germany, ≥99.8% purity) solvent and ultrasonicated (Elmasonic P 00037510, 30 kHz, Singen, Germany) for 3 h. The micro flakes of NG were separated by centrifugation at 2500 rpm (Frontier Centrifuge FC5706, Parsippany, NJ, USA), collected, washed by ethanol, and dried under vacuum at 80 °C for 12 h. The ultrasonicated highly oriented pyrolytic graphite (HOPG) (NT-MDT, Moscow, Russia) was prepared using the same method. The pristine NG sample is further referred to as “NG” and the ultrasonicated NG and HOPG samples are referred to as “US-NG” and “US-HOPG”.

### 2.2. Electrode Preparation

The graphite electrodes were prepared by casting a slurry onto a current collector of polytetrafluoroethylene (PTFE)-treated hydrophobic carbon paper of 150–200 μm thickness (120 Toray Carbon paper, Fuel Cell Store, College Station, TX, USA). The slurry was prepared by manually mixing the US-NG with conducting carbon black (Super-P, Imerys, Paris, France), polyvinylidene difluoride (PVDF) (Kynar^®^ HSV 1800, Arkema, Colombes, France) as a binder, and NMP (75:15:10 wt. ratio) in a mortar. The as-prepared electrodes were dried at 80 °C under vacuum in an oven for 12 h prior to electrochemical testing. 

### 2.3. Electrolyte Preparation

The 2.4 M Al(ClO_4_)_3_ aqueous electrolyte solution was prepared by dissolving 29.24 g of Al(ClO_4_)_3_.9H_2_O (Alfa Aesar, Haverhill, MA, USA) in 10 mL of deionized (DI) water. The solution was stirred for 30 min at room temperature to obtain a clear solution.

### 2.4. Electrochemical Cycling

The galvanostatic charge/discharge cycling was performed from −0.08 to 1.55 V (vs. Ag/AgCl) in a three-electrode standard electrochemical cell by using an Ag/AgCl pseudo-reference electrode and a platinum wire as a counter electrode in the concentrated Al(ClO_4_)_3_ aqueous electrolyte. A µ-Autolab type III workstation (Metrohm, Herisau, Switzerland) was used for electrochemical measurements. Cyclic voltammetry (CV) tests were conducted in the potential window of −0.08 to 1.58 V (vs. Ag/AgCl) at various sweep rates from 1 to 9 mVs^−1^. For the ex situ characterizations, all the samples were washed in DI water and dried in a vacuum oven at 70 °C for 2 h prior to the measurements. The electrodes were discharged to −0.08 V and charged till 1.55 V (vs. Ag/AgCl), and subsequently referred to as “discharged” and “charged”, respectively, and those without any electrochemical activity were referred to as “fresh”.

### 2.5. Material Characterization

Scanning electron microscopy (SEM) images were obtained using a TESCAN MAIA3 microscope (Brno, Czech Republic). X-ray diffraction (XRD) was performed on Bruker D8 (Bruker, Karlsruhe, Germany) with Cu Kα radiation (λ = 1.5418 Å). The X-ray diffractogram was recorded at 0.01° per step with a slit width of 0.6 mm in order to improve the accuracy of the XRD measurement. A Horiba Lab-RAM HR (Lille, France) equipped with an Olympus microscope, a 633 nm He-Ne excitation laser operated at 1 mW power by 600 lines/mm grating (1.8 cm^−1^ point-to-point spectral resolution), and a 100× long working distance objective, was used to record the Raman spectra. 

The in situ Raman spectroelectrochemistry measurements were performed in a home-made cell, using the same setup with the potential held for about 1000 s for each step. The prepared samples served as the working electrodes, a chlorinated Ag wire was used as the reference electrode, and Pt was used as the counter electrode; Autolab PGSTAT30 (Metrohm, Herisau, Switzerland) was used for the Raman-SEC measurement. The operando XRD analysis was performed using the same setup and conditions as the ex situ analyses. The X-rays penetrate through the KAPTON tape (Tob Xiamen, Xiamen, China) window from the top. The XRD was connected with a battery tester (Neware, Hong Kong, China) for galvanostatic charge/discharge in a two-electrode setup at 20 μAg^−1^ current density. All the measurements were recorded at 2*θ* in the range of 10–60° with a step of 0.02° using a *Lynxeye XE-T* detector and applying a voltage of 40 kV and 30 mA current.

The ex situ XPS of the fresh, charged, and discharged samples were performed using an AXIS Supra photoelectron spectrometer (Kratos Analytical Ltd., Manchester, UK) with an Al Kα monochromatic energy source. The XPS analysis of US-NG material was performed by ADES 400 (VG Scientific, London, UK) with an Al Kα monochromatic energy source. The spectra were recorded after sputter-etching the samples with Ar^+^ ion clusters (the number of Ar atoms in the cluster was 1000) of 5 keV energy for 10 min. The elemental composition was calculated from the high-resolution core level spectra with respect to the relative sensitivity factor. The determined uncertainties in concentration correspond to the deviation in the measured atomic % of the elements, and were calculated by ESCApe 1.4.0 software (Kratos analytical Ltd., Manchester, UK).

## 3. Results and Discussion

### 3.1. Structural Properties of Graphite and Electrochemical (De)Intercalation of ClO_4_^−^ Anion

The SEM images of the US-NG (Appendix A in the Appendix A) show more disjointed and crumpled structures in comparison to NG (Appendix A). The ultrasound treatment of graphite leads to a partial disintegration of the bulk crystallites to flakes of varying thickness and lateral dimensions. Figure 1a displays the XRD patterns of the NG and US-NG. The resulting structural parameters are summarized in Table 1. The XRD (002) peaks of the NG and US-NG were recorded at 2*θ* of 26.53(8)° and 26.54(4)°, corresponding to the interlayer spacing (*d*_002_) of 3.35(6) Å and 3.35(5) Å, respectively. For comparison, the XRD (002) peak of the US-HOPG (Appendix A) recorded at 2*θ* of 26.53(1)° corresponds to the interlayer spacing of 3.35(6) Å. The degree of graphitization (*g*), calculated on the basis of *d*_002_ (see Appendix A) was also found to be very similar, and close to perfectly crystalline graphite for NG, US-NG, (Table 1) and US-HOPG (Appendix A). The coherent domain size (*L*_c_) for the stacking along the c-axis for the hexagonal graphitic structure was calculated using the Scherrer equation (Appendix A) [40]. The calculated *L*_c_ was 52.20 nm to 84.89 nm for NG and US-NG, respectively (Table 1), and 97.50 nm for the US-HOPG (Appendix A). 

Figure 1b shows the Raman spectra of NG and US-NG. The Raman G band at 1583 cm^−1^ and the 2D band at 2500–2700 cm^−1^ are the typical Raman signatures of graphite [41]. In addition, disorder-related D and D′ bands at 1350 cm^−1^ and 1620 cm^−1^, respectively, can be seen in all the spectra. Importantly, the ultrasonication results in higher intensities of the D and D′ bands, thereby pointing to a more defective crystalline structure (Figure 1b). The inter-defect distance (*L_D_*), defect density (n*_D_*) and lateral domain size (*L*_a_) were calculated from the ratio of the D and G band intensities (*I_D_/I_G_*) by the equations as reported by Cancado et al. (Appendix A) [42]. The calculated structural parameters are summarized in Table 1. The larger *I_D_/I_G_* ratio of the US-NG (*I_D_/I_G_* = 0.18) corresponds to lattice changes with a smaller inter-defect distance (*L_D_*) of ~ 44 nm in comparison to the NG (*I_D_/I_G_* = 0.11, *L_D_* ~ 51 nm). The other two structural parameters, defect density (n*_D_*) and crystallite size (*L*_a_), follow the same trend as *L_D_* (Table 1). The US-HOPG (*I_D_/I_G_* = 0.04) (Appendix A) shows smaller lattice changes with an *L_D_* of ~84 nm (Appendix A). *L*_a_ refers to the mean lateral crystallite size, an important parameter to describe the structural properties of the graphite. As can be seen from the data above, the ultrasonication treatment of NMP does not have a substantial influence on the interlayer spacing (and graphitization degree). However, the coherent domain size *L*_c_ of NG increases with the treatment. The increase in *L*_c_ might be caused by the relaxation of the carbon hexagons as the lateral dimensions of the graphitic planes become smaller. The ultrasound cleaves the layers at the weaker defective sites, which might have been imposing stress, thereby lowering the *L*_c_ in the pristine material. Similar behavior was observed for carbon fibers [43]. Moreover, the increase in *I_D_/I_G_* after ultrasonication corresponds to more functional groups’ decorated defects at the edges of graphite, which in turn provide more active sites and enhance the electrochemical activity [44]. The Raman spectra-derived parameters corroborate the data obtained by XRD, evidencing the increase in structural damage caused by the ultrasonication treatment which has, consequently, a substantial influence on electrochemical performance and will be discussed in detail in the following sections. 

The electrochemical characteristics of US-NG were studied in 2.4 M Al(ClO_4_)_3_ aqueous electrolyte concerning the (de)intercalation of the ClO_4_^−^ anion. Figure 2a shows the cyclic voltammograms (CVs) of NG and US-NG recorded at a sweep rate of 1 mVs^−1^. The CVs show two redox peaks in both the cathodic and anodic regions. The electrochemical intercalation of the ClO_4_^−^ ions in the graphite is attributed to the two oxidation peaks at 1.43 V (O^⸍^_A_) and 1.55 V (O_A_) (~4.7 to 4.8 V vs. Li/Li^+^) and the reduction peaks at 1.16 V (R^⸍^_C_) and 1.35 V (R _C_) [45,46]. The different multiple peaks during the intercalation/deintercalation of the anion into graphite correspond to the different staging mechanisms [47,48]. The visibleble plateaus at around 1.55 V during galvanostatic charge, and at 1.35 V and 1.16 V during the discharge of NG and US-NG (Figure 2b), correspond to the intercalation and (de)intercalation of ClO_4_^−^ ions, respectively, which is in line with the CV profile. Likewise, the CV of US-HOPG (Appendix A) shows similar redox peaks (O_A_, O^⸍^_A_, R_C,_ and R^⸍^_C_). The difference in the galvanostatic charge/discharge profile of NG and US-NG (Figure 2b) could be due to the relatively smaller lateral crystallite size of the US-NG compared to NG. 

To reveal the nature of the electrochemical process of US-NG in detail, the analysis of CVs with varying scan rates (Figure 2c) can unbraid the contributions of diffusion and surface-controlled processes by using the power–law relationship of the peak current (*i*) and scan rate (*v*) (SI Equation (S6)) [49,50]. Figure 2d shows the CV kinetics of US-NG, where the fitted *b* values of 0.74 for the oxidation peak O_A_ and 0.91 for the reduction peak R_C_ indicate the contribution of both ionic diffusion and surface-controlled processes, while the *b* value of 1.0 for the R^⸍^_C_ reduction peak evidences the contribution of purely surface-controlled processes.

The galvanostatic charge/discharge cycling curves of US-NG were recorded in 2.4 M Al(ClO_4_)_3_ aqueous electrolyte at a current density of 250 mAg^−1^. The US-NG shows a stable discharge capacity of ~18 mAhg^−1^ for 300 cycles, with Coulombic efficiency larger than 80% (Figure 2e). The discharge capacity of ~18 mAhg^−1^ for US-HOPG in 2.4 M Al(ClO_4_)_3_ aqueous electrolyte at a current density of 250 mAg^−1^ was also observed for 125 cycles, as shown in Appendix A. A similar charge/discharge performance for graphitic electrode materials in both aqueous and non-aqueous electrolytes has been reported previously [23,39,51].

Ultrasound treatment leads to smaller crystallites (larger surface) and more active sites at the surface, which enhances electrochemical activity [52]. The smaller lateral crystallite size might have a substantial influence on the cyclic stability, as well as on the reversibility of the (de)intercalation of the ClO_4_^−^ anion in US-NG. It is well known that defects are introduced into the lattice structure of graphite due to the attachment of functional groups during the ultrasound treatment in different solvents [53,54,55]. Skaltsas et al. reported that the ultrasonication of graphite in NMP induces the formation of defects accompanied by the increase in the concentration of the oxygenated species (COOH, COO) in the material [53]. Here, we observed the presence of the hydroxyl and carboxyl oxygen-containing functional groups attached to the graphitic carbon of US-NG by XPS analysis, as shown in Appendix A. We note that the analysis was performed after Ar^+^ ion etching to minimize the potential influence of airborne contamination. The elemental analysis for US-NG is presented in Appendix A. The oxygenated species at the graphite lattice could originate from the decomposition of NMP or surface oxidation during the ultrasound treatment [26]. The functional groups decorating the graphite surface could facilitate the large-size ions accessing the graphite interlayer galleries through surface adsorption during the electrochemical process [56,57,58,59,60]. Figure 2f displays the specific capacity of US-NG recorded at various current densities from 100 mAg^−1^ to 500 mAg^−1^. It also shows a stable electrochemical performance with an increase in Coulombic efficiency at high current density (500 mAg^−1^). Appendix A shows the rate capability of US-HOPG from 100 mAg^−1^ to 500 mAg^−1^.

### 3.2. In Situ Characterization

In order to obtain deeper insights on the ionic (de)intercalation into US-NG, in situ Raman SEC was performed. Figure 3a displays the Raman spectra of US-NG acquired in situ during the charge/discharge cycle. The G peak, being the most sensitive to the interaction of ions with the graphite planes, splits into two components. The low-frequency (~1584 cm^−1^) G(*i*) mode and high-frequency (~1613 cm^−1^) G(*b*) mode appear due to the formation of GICs with the anions inside the interlayer galleries [25,61,62]. The splitting of the G peak at 1.40 V (vs. Ag/AgCl) during the charging process, accompanied by a further intensity increase in the G(*b*) peak, corresponds to the gradual intercalation of the ClO_4_^−^ ion in the interlayer spaces of graphite. The complete disappearance of the G(*b*) mode during the discharge process is attributed to a fully reversible deintercalation [63,64,65]. This observation is in line with the stable discharge capacity in the case of US-NG over long cycles, as shown in Figure 2e. In spite of the hysteresis in the G peak position during the charge/discharge cycle, Figure 3b shows that the graphite lattice is completely restored after the (de)intercalation of the ClO_4_^−^ anion. Comparatively, during a galvanostatic discharge, in the case of untreated NG, the persistence of the G(*b*) mode during the discharge process was attributed to the only partial reversibility of the (de)intercalation of the ClO_4_^−^ anion, as shown in our previous publication [28].

Operando XRD proved to be an efficient tool to monitor the structural lattice modifications taking place alongside the GIC formation during the charge/discharge process [66,67]. Figure 4 shows the evolution of the (002) and (004) reflections of US-NG during one charge/discharge cycle. The characteristic graphitic (002) reflection broadens and splits during the charge process due to the intercalation of the ClO_4_^−^ anion [23]. The emerged reflection has 2*θ* of ~25.5°, corresponding to *d*_002_ of ~3.49 Å. Analogous behavior can be observed in the (004) reflection. Decreases in broadening and intensity in the main reflections were observed upon charging, which also depicts the ionic intercalation into graphite. The peaks were fully recovered during the discharge process (Figure 4). 

### 3.3. Ex Situ Spectroscopic Characterization

To describe the (de)intercalation process in more detail, additional ex situ spectroscopic techniques, such as ex situ Raman spectroscopy and ex situ XPS, were performed on the fresh, charged, and discharged US-NG electrodes. 

The ex situ Raman spectra of US-NG show an increase in the intensity ratio of the D to G peaks (*I*_D_/*I*_G_) during the charged state (0.31) in comparison to the discharged (0.24) and fresh (0.17) states (Appendix A), illustrating the reversible structural changes during the intercalation of the ClO_4_^−^ anion [39]. Similarly, the ex situ Raman spectra of US-HOPG (Appendix A) also show an increase in *I*_D_/*I*_G_ ratio (0.17) during charging in comparison to the discharged (0.15) and fresh (0.03) states. The incomplete recovery of the small D peak intensity points to the partial irreversibility of the intercalation in US-HOPG.

The chemical composition of fresh, discharged, and charged US-NG electrode materials were analyzed by XPS (Figure 5). Figure 5a shows the survey XPS spectra of the electrodes, with the lines assigned to Cl, Al, O, C, and F labelled. The presence of F comes from the electrode binder [68]. Figure 5c,d shows the high-resolution Cl 2p core-level XPS spectra of US-NG during the charged/discharged state, demonstrating the presence of the two chemical states of Cl. The lower-energy component at ~200 eV corresponds to Cl covalently bonded to carbon atoms (C-Cl) [69]; the peaks at 201.9 eV and 200.1 eV belong to the 2p_1/2_ and 2p_3/2_ energy levels, respectively. The higher-energy state at ~208 eV corresponds to ClO_4_^−^. The peaks at 209.6 eV and 208.0 eV correspond to Cl 2p_1/2_ and Cl 2p_3/2_, respectively, attributed to the adsorbed ClO_4_^−^, possibly originating from the electrolyte presence at the surface of the electrode [70,71]. Cl 2p_1/2_ and Cl 2p_3/2_ at 208.0 eV and 206.4 eV, respectively, reveal the intercalated ClO_4_^−^ ion [14,72]. The elemental composition (Appendix A) and selected ratios of the atoms or ions (Table 2) evidence that higher amounts of ClO_4_^−^ were present in the charged than in the discharged US-NG. Importantly, the ratio of intercalated/adsorbed ClO_4_^−^ was observed to be significantly higher for the charged state than for the discharged state (Figure 5c,d). We noted that an exact quantification of the intercalated/adsorbed ClO_4_^−^ is not possible due to the overlap of the spin-split peaks at the binding energy of 208.0 eV (Cl 2p_1/2_ of the intercalated ClO_4_^−^ and Cl 2p_3/2_ of the adsorbed ClO_4_^−^). Nevertheless, the amount of intercalated ClO_4_^−^ is very low in the discharged state, also taking into account the very low Cl content (Appendix A), which can explain the structural reversibility seen in Raman SEC and XRD. Figure 5b shows the Al 2p core level at 75.5 eV binding energy. The concentration of Al (Appendix A) was observed to be higher for the charged US-NG electrodes, compared to the discharged one. Additionally, the Al to C ratio in the charged US-NG is higher than in the discharged US-NG (Table 2), which might correspond to the trapped electrolyte in the US-NG electrode.

## 4. Conclusions

The effect of the ultrasound treatment of natural and highly oriented pyrolytic graphite on the reversibility of electrochemical (de)intercalation of the ClO_4_^−^ anion was studied in aqueous Al(ClO_4_)_3_ electrolyte solution. The ultrasonication of graphite in NMP induced the formation of defects accompanied by a reduction in its crystallite size, *L*_a_. The smaller *L*_a_ of US-NG in comparison to pristine NG facilitated the easier escape of the large-size ions from the graphite interlayer galleries, thereby enhancing the reversibility and cyclic stability. The in situ Raman SEC and operando XRD of US-NG confirmed the reversibility of the structural changes in the graphite induced by the ClO_4_^−^ intercalation, although a remaining signal of the intercalated ClO_4_^−^ species was detected by ex situ XPS. The study shows that ultrasound treatment offers a practical approach for improving the reversibility of electrochemical intercalation processes in graphite, which can also help in designing novel functional graphite intercalation materials for energy storage systems.

## Figures and Tables

**Figure 1 nanomaterials-12-03932-f001:**
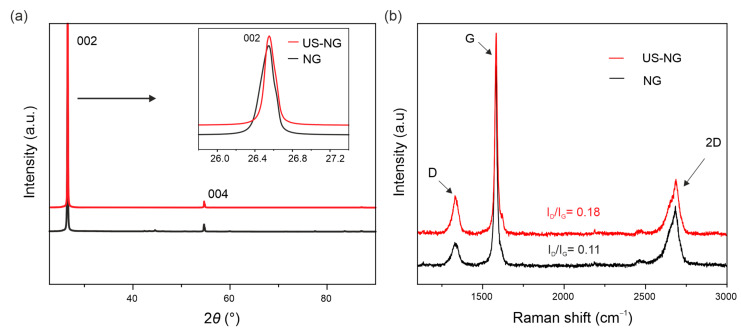
Structural characterization: (**a**) XRD and (**b**) Raman spectra of NG (black curves) and US-NG (red curves).

**Figure 2 nanomaterials-12-03932-f002:**
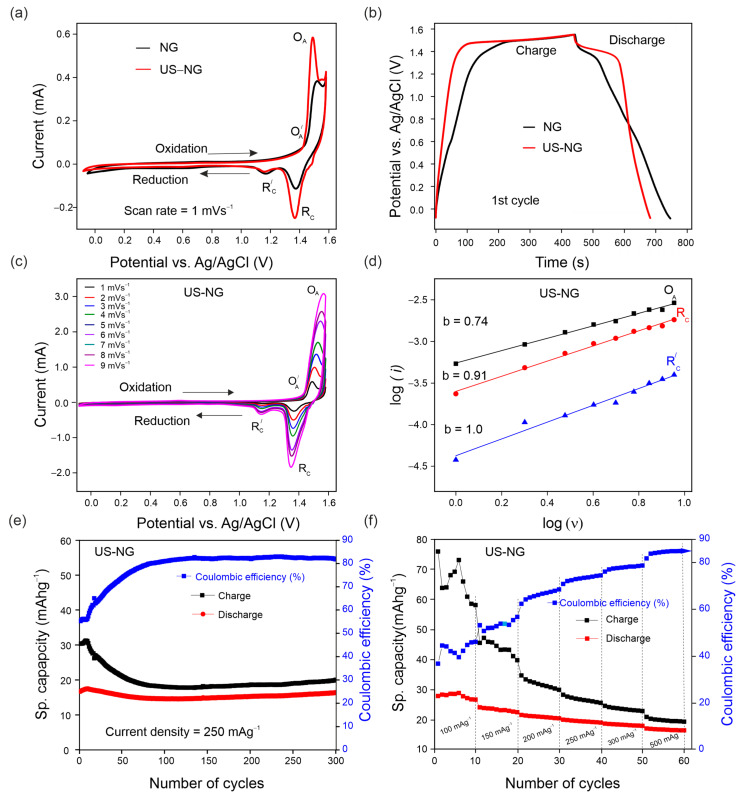
Electrochemical performance of US-NG using 2.4 M Al(ClO_4_)_3_ aqueous electrolyte solution: (**a**) CVs of NG and US-NG at 1 mVs^−1^; (**b**) galvanostatic charge/discharge potential profile of NG and US-NG at a current density of 250 mAg^−1^ (1st cycle); (**c**) CVs of US-NG at different scan rates (1 to 9 mVs^−1^); (**d**) electrochemical kinetic of log (*i*) vs. log (*ν*) at the redox peaks for US-NG; (**e**) galvanostatic cycling charge/discharge performance of US-NG obtained at a current density of 250 mAg^−1^; (**f**) the rate capability of US-NG for specific capacities at various current densities ranging from 100 mAg^−1^ to 500 mAg^−1^ using 2.4 M Al(ClO_4_)_3_ aqueous electrolyte solution.

**Figure 3 nanomaterials-12-03932-f003:**
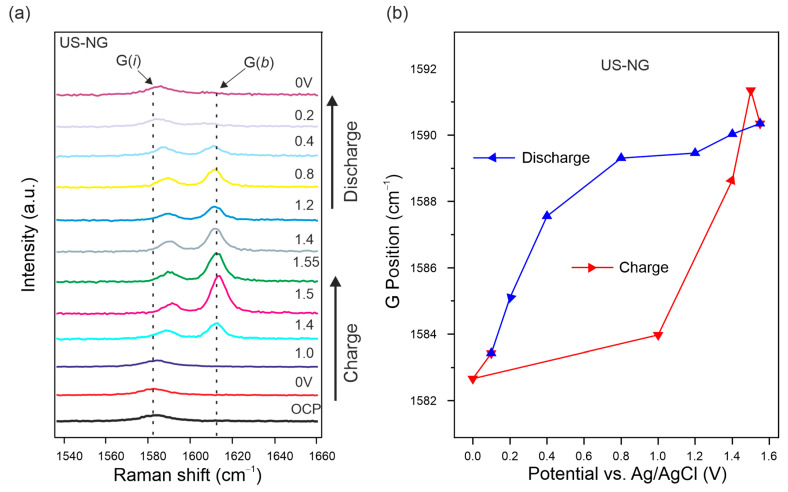
In situ Raman SEC of US-NG during charge/discharge cycle in 2.4 M Al(ClO_4_)_3_ aqueous electrolyte. (**a**) Evolution of the Raman spectra in the G peak spectral region. (**b**) Raman G peak position evolution fitted as one Lorentzian line shape. The applied potentials were held for 1000 s for each step.

**Figure 4 nanomaterials-12-03932-f004:**
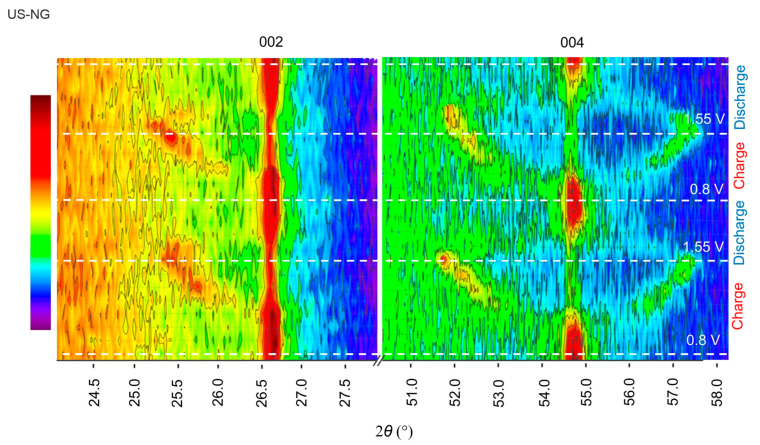
Operando XRD analysis of US-NG during charge/discharge process in 2.4 M Al(ClO_4_)_3_ aqueous electrolyte solution.

**Figure 5 nanomaterials-12-03932-f005:**
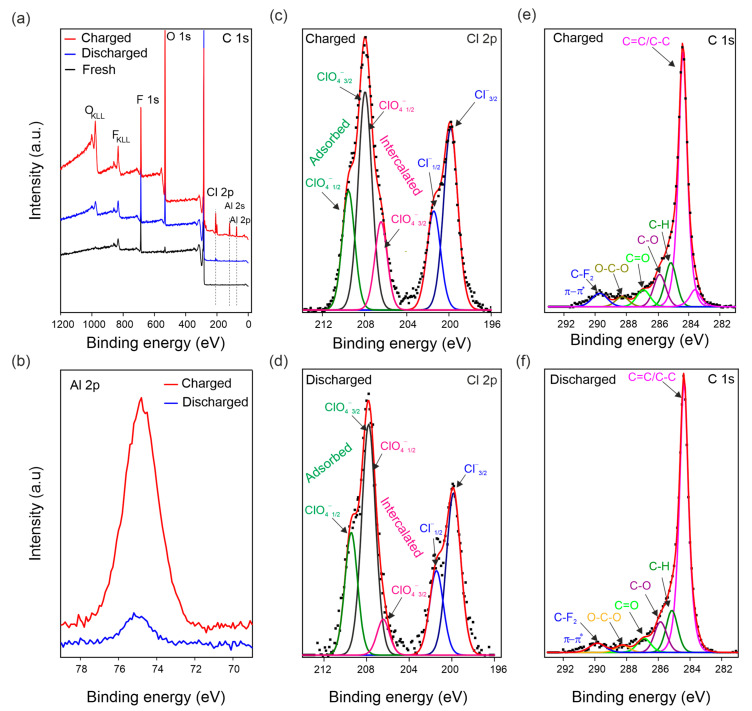
Ex situ XPS analysis of US-NG in fresh, charged, and discharged states using 2.4 M Al(ClO_4_)_3_ aqueous electrolyte solution. (**a**) Survey XPS spectrum; (**b**) high-resolution XPS spectra of the Al 2p core levels in the charged and discharged states; deconvoluted high-resolution XPS spectra of the Cl 2p core levels in the charged (**c**) and discharged states (**d**) (note, the Cl 2p_1/2_ of intercalated ClO_4_^−^ and Cl 2p_3/2_ of adsorbed ClO_4_^−^ overlap at 208.0 eV); deconvoluted high-resolution XPS spectra of the C 1s core levels in the charged (**e**) and discharged states (**f**).

**Table 1 nanomaterials-12-03932-t001:** Quantification of structural parameters of NG and US-NG from XRD and Raman spectroscopy.

Material	*g* (%)	*L*_c_ (nm)	*L*_D_ (nm)	*n*_D_ (cm^−2^)	*L*_a_ (nm)
NG	97.67	52.20	51.26 ± 0.14	(12.33 ± 0.27) × 10^9^	350.29
US-NG	98.58	84.89	44.43 ± 0.02	(20.20 ± 0.11) × 10^9^	214.07

**Table 2 nanomaterials-12-03932-t002:** Concentration ratios of ClO_4_^−^ and Cl and Al and C for US-NG derived from XPS.

Sample	US-NG	
	Charged	Discharged
Cl/C	0.059	0.010
ClO_4_^−^/C	0.335	0.080
Al/ClO_4_^−^	0.227	0.080
Al/C	0.076	0.010

## Data Availability

The data presented in this study are available on request from the corresponding author. The data are not publicly available due to IP protection revision of related studies.

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
