# Peer review of "The Effects of Ultrasound Treatment of Graphite on the Reversibility of the (De)Intercalation of an Anion from Aqueous Electrolyte Solution"

_nanomaterials, 2022, doi:10.3390/nano12223932_

Round 1

Reviewer 1 Report

The paper reports the ultrasound treatment of graphite on the reversibility of (de)intercalation of anion from aqueous electrolyte solution. The paper was well organized and presented. However, several technical issues shall be cleared to ensure the publication of the paper so as to stir interests among different audiences:

1) The paper mentioned the reduction of the crystalline size in many places throughout the manuscript. As estimated from XRD and Raman, the crystalline size reduced from 51 nm to 44 nm. However, is it more clear to provide TEM/SEM microscopic image? It is not convincing if the size reduction could be dramatic?

2) Regarding the XRD in Fig.1(a), the diffraction peak (002) is compared in the inset. But the peak shape distorted in US-NG, is it due to artefacts from the measurement? The angular resolution of the XRD measurement shall be emphasized?

3) As a surface sensitive technique, is there particular effects in XPS in probing nanocrystalline system? In other words, what are the uncertainties of the concentration ratio determined from XPS?

Reviewer 2 Report

This manuscript reports the preparation of the ultrasound-treated graphite and its application for the reversibility of ClO4- anions in the aqueous-based electrolyte. Compared to pristine graphite, ultrasound-treated US-NG has different properties including considerable defects formation and reduction of the crystallite size, which led to the facile reversible (de)intercalation of ClO4- anions from/to US-NG. Several characterizations were performed, which provide solid evidences. Therefore, the reviewer recommends this manuscript to publish in Nanomaterials after addressing following points.

1.     Highly concentrated electrolyte can widen the electrochemical stability windows. The authors only use the 2.4 M Al(ClO4)3 electrolyte. Did the authors try to increase the concentration of the electrolytes?  

2.     The high-resolution XPS analysis of the pristine US-NG should be also provided and discussed.

3.     The authors claimed that the increased defect sites are beneficial for the electrochemical performance. However, it has been also known that if the structure has higher defects, the conductivity of the electrodes decreased, leading to inferior electrochemical properties. Therefore, the reviewer suggests to discuss on the advantageous effects of the defects; how the functional groups in the defects act as supporters to enhance the reversibility of the anions.

4.     Also, if it helps reversible (de)intercalation of anions, the reviewer suggests that authors provide the (de)intercalation mechanisms during charge and discharge.

5.     The supplementary information has not been provided.

Round 2

Reviewer 1 Report

The authors has addressed all my questions and answered satisfactorily. I would recommend to accept this paper.

Reviewer 2 Report

The authors revised well the manuscript based on the reviewer's comment. Therefore, the manuscript is ready to be published in Nanomaterials.